# Carbonaceous Materials Porosity Investigation in a Wet State by Low-Field NMR Relaxometry

**DOI:** 10.3390/ma15249021

**Published:** 2022-12-16

**Authors:** Eva Kinnertová, Václav Slovák, Tomáš Zelenka, Cyril Vaulot, Luc Delmotte

**Affiliations:** 1Department of Chemistry, Faculty of Science, University of Ostrava, 30. Dubna 22, 701 03 Ostrava, Czech Republic; 2Institut de Science des Matériaux de Mulhouse (IS2M), CNRS UMR 7361, Université de Haute-Alsace, 15 Rue Jean Starcky, 68057 Mulhouse, France

**Keywords:** porosity, low-field NMR relaxometry, relaxation time, mesopores, relaxivity

## Abstract

The porosity of differently wetted carbonaceous material with disordered mesoporosity was investigated using low-field ^1^H NMR relaxometry. Spin–spin relaxation (relaxation time *T*_2_) was measured using the CPMG pulse sequence. We present a non-linear optimization method for the conversion of relaxation curves to the distribution of relaxation times by using non-specialized software. Our procedure consists of searching for the number of components, relaxation times, and their amplitudes, related to different types of hydrogen nuclei in the sample wetted with different amounts of water (different water-to-carbon ratio). We found that a maximum of five components with different relaxation times was sufficient to describe the observed relaxation. The individual components were attributed to a tightly bounded surface water layer (*T*_2_ up to 2 ms), water in small pores especially supermicropores (2 < *T*_2_ < 7 ms), mesopores (7 < *T*_2_ < 20 ms), water in large cavities between particles (20–1500 ms), and bulk water surrounding the materials (*T*_2_ > 1500 ms). To recalculate the distribution of relaxation times to the pore size distribution, we calculated the surface relaxivity based on the results provided by additional characterization techniques, such as thermoporometry (TPM) and N_2_/−196 °C physisorption.

## 1. Introduction

In recent decades, considerable attention has been paid to porous carbonaceous materials, due to their wide spectrum of use, such as adsorbents, catalyst supports, gas storage/separation media, electrodes for supercapacitor, advanced materials for electronic applications, and many others [1]. According to the countless published research articles on carbonaceous materials, their porosity in connection with pore size and surface area is one of their key characteristics and one of the most studied fields. However, the possible pore size variations with the change from the dry state to the wet state remain a research field that is not entirely investigated.

Many natural and synthesized porous media are composed of combined (hierarchical) porosities: a microporosity (pore width < 2 nm), where the adsorbate is trapped as an adsorbed phase due to the high surface area, and a required mesoporosity (2–50 nm) connected to a macroporosity (>50 nm), in order to ensure the fast transport of the adsorbate to the micropores [2,3]. Today, there are various methods for characterising the porosity, and one of the most common methods is gas physisorption based on the physical adsorption of gases under controlled conditions of temperature and pressure. The physisorption is performed on perfectly dried and degassed samples to remove any potential moisture that could interfere with the analysis. However, there are some issues related to the effective sample outgassing procedure: (i) the evacuation of the finely powdered sample carries the risk of its elutriation, and (ii) the vacuum itself, together with the high outgassing temperature, can also cause a significant structural change in the sample in the context of its surface chemistry, porosity, and volumetric changes (deformations) [4,5]. Adsorbent elastic deformation can be observed for a wide range of porous materials, such as activated carbon, charcoal, aerogels, porous glass, porous silicon, etc., and as irreversible structural changes for metal organic framework materials. During the final stage of the drying procedure, the phase (water) entrapped in nanocavities, in the form of a few atomic or molecular layers, exhibits a high surface-to-volume ratio with strong surface interactions. This phenomenon leads to the formation of considerable forces affecting the walls of microporous systems, which cause the deformation of all pore sizes or even macroscopic changes in the case of less rigid structures, such as organic materials [2,6]. Such sample deformations do not represent its original (natural) state and can be minimized by using more mild conditions, but at the expense of prolonged outgassing procedure and the risk of an insufficiently dried sample. In addition, adsorbent deformation may also occur by exposing it to a high vacuum, which is required for its proper drying, but also during the measurement itself. Despite the fact that gas adsorption is routinely used for the surface and textural characterization of porous materials, because it allows the assessment of a wide range of pore sizes, including the complete range of micro- and mesopores (up to about 50 nm), it is necessary to take into account that, in the ideal case gas physisorption, the measurements characterize the sample surface properties in an absolutely dry state. This can be far from their properties in real applications, such as adsorption in aqueous media. 

The restriction related to sufficient drying procedures (evacuation at elevated temperature) can be bypassed by using techniques that allow for the characterization of the porosity of the materials in the wet state at ambient pressure. Thermoporometry (TPM), which is performed through DSC measurement, is one of them. TPM is based on the decrease of the melting/freezing temperature of a solid/liquid phase entrapped in the pores, which is caused by the small size of crystals formed inside the pores [7]. The method is applicable for mesopores and small macropores [8,9], as there is a limitation for the evaluation of microporous materials, since no phase transition (melting/freezing) takes place in such small pores. The porous properties of ordered mesoporous silicas (especially SBA-15, MCM-41) [10,11] and disordered ones [12,13] are often studied using TPM. Although TPM has considerable advantages over gas adsorption, this method is not as widely used, probably due, among other things, to the ambiguous constants required to calculate the pore size distribution [14]. As already mentioned, the advantage of this technique lies in its ability to analyse samples in their initial (wet) state. Taking into account the similar principle, the technique of NMR cryoporometry is closely related to TPM; however, it is more often used than TPM. 

In general, NMR methods have become an alternative approach to the conventional gas physisorption technique for the study of both microporous and mesoporous materials and the surface interactions of confined liquids [15,16]. Low-field ^1^H NMR relaxation is used for the simple and fast measurement of wetted materials. The first use of this technique for characterization of the pore structure of porous media, primarily in the oil industry, can be dated to around 1950 [17]. In recent years, the interest of this powerful tool has increased, and a combination of several NMR techniques (relaxometry, cryoporometry, or diffusometry) is especially useful, as it allows elucidation of a multitude of important characteristics of different types of porous materials [16]. NMR relaxometry can be used for both the investigation of molecular dynamics of systems during the synthesis processes [18,19] and for the final prepared porous materials [20,21,22]. The technique is conveniently used for the characterization of water in rock and soils samples [23,24,25,26,27], in combination with NMR cryoporometry or diffusometry. It provides information on the porosity of various carbonaceous materials, such as xerogels [28], coals [29], mesoporous carbon [15], other carbons [30], and also of silica materials, such as MCM-41 and SBA-15 [31]. The fact that the speed of relaxation of magnetized nuclei depends on the mobility of molecules is the basis for NMR relaxometry usage in the field of porous material characterization. The analysis relies on finding the number of components connected with different types of protons present in the studied system (identification of proton populations [32]) through relaxation time *T*_1_ (longitudinal relaxation) or *T*_2_ (transverse relaxation). The appropriate parameters, amplitude, and relaxation time corresponding with given types of protons are optimized and can be used for the characterization of the pore size distribution of various porous materials. The evaluation of the relaxation times is carried out in the major part of investigations using special software, and the literature often does not provide important or more detailed information about this procedure.

The aim of this work is to show a simple conversion of relaxation curves to an estimate of the distribution of relaxation times, without the necessity of special software. Based on the obtained distribution of relaxation times, it is possible to describe the evolution of *T*_2_ populations for different pore filling ratio and correlate one of the *T*_2_ populations with the mesopore diameter of the material. In addition, the results also enabled the calculation of the surface relaxivity of carbonaceous materials on the TPM and nitrogen adsorption data.

## 2. Materials and Methods

### 2.1. Preparation of the Mesoporous Carbon Material

The preparation of micro and mesoporous carbon was based on cross-linking of phloroglucinol (Sigma Aldrich, Saint Louis, MO, USA, >99%) and glyoxal (40% aqueous solution) under acidic conditions in the presence of triblock copolymer Pluronic F127 (Sigma Aldrich) as a structure-directing agent [33]. Phloroglucinol (3.3 g) and Pluronic F127 (6.5 g) were dissolved in a mixture of 162 mL of absolute ethanol (p.a., 99.8%) and 1.2 mL of HCl (p.a., Mach Chemikálie s.r.o., Slezská Ostrava, Czech Republic, 37%). After complete dissolution at room temperature, 3.25 mL of aqueous solution was added. The mixture was stirred for 1 h and aged at room temperature for four days to the appearance of a macroscopic phase separation. The upper layer (mainly formed by the mixture of water and ethanol) was removed, and the lower layer, containing the yellow polymer-rich gel, was poured into a Petri dish and thermo-cured at 80 °C overnight. The sample was pyrolyzed in the atmosphere (30 min at 100 °C, 2 h at 400 °C, heating rate 10 °C/min). The prepared mesoporous carbon was crushed and denoted as MC.

### 2.2. Thermoporometry Analysis

The presence of mesopores in MC sample was evidenced by thermoporometry using DSC 1 Star System (Mettler Toledo, Columbus, OH, USA), with demineralized water as a liquid probe. Based on preliminary experiments, the mass ratio 1:2 of sample-to-water was chosen. Five mg of sample was placed in an aluminium pan, gently compressed, and then 10 μL of demineralized water was added. The pan was hermetically sealed and left to stabilize at room temperature for 1 h before analysis. The procedure of pre-freezing (to ensure a complete freezing of the liquid present in the pores) was according to the following program: cooling to −90 °C (5 °C/min), heating to −0.3 °C (5 °C/min), 10 min at −0.3 °C, cooling to −90 °C (5 °C/min), 5 min at −90 °C. The thermoporometry measurement was then performed by heating to 25 °C, with a heating rate of 5 °C/min. The baseline of DSC melting peak of water in pores and onset temperature of a bulk water melting peak were determined for the differential pore size distribution (PSD) calculations. The pore volume was calculated by integration of the PSD in the given range of pore sizes. Additional information can be found in Ref. [7].

### 2.3. Adsorption of Nitrogen

The porosity characteristics of the MC sample were determined by nitrogen adsorption–desorption at −196 °C using a manometric method (Autosorb iQ-XR, Quantachrome). Before analysis, the sample was outgassed in a three-step program at the following temperature and time period: 65 °C for 30 min, 105 °C for 30 min, and 300 °C for 600 min, with a heating rate of 3 °C/min. The adsorption and desorption isotherms were obtained in the range of equilibrium relative pressures 10^−7^–0.995. The volume and surface of ultramicropores (*V_ultramicro_*, *S_ultramicro_*), supermicropores (*V_supermicro_*, *S_supermicro_*), and mesopores (*V_meso_*, *S_meso_*) were calculated by integration of the pore size distribution (PSD) in the given range of pore sizes. The PSDs were obtained by fitting experimental data with a QSDFT adsorption kernel (a set of the theoretical isotherms; ASiQwin software, Quantachrome), assuming a heterogeneous carbon surface of slit-shaped micropores and cylindrical mesopores.

### 2.4. H-NMR Relaxometry Analysis

All NMR measurements were performed using the low-field NMR Spectrometer Minispec mq 20 (Bruker, Rheinstetten, Germany), operating at a proton resonance frequency and a magnetic field strength of 0.47 T. Spin–spin relaxation *T*_2_ (transverse relaxation) of ^1^H was determined by the Carr-Purcell-Malboom-Gill (CPMG) pulse sequence. Taking into account the rigid nature of the sample, the analyses were realized with the minimal possible time delay, τ = 0.05 ms. In the case of samples wetted by the procedure described below (Figure 1), an additional, longer time τ was used (τ = 0.2 ms), in order to observe total decay, as there was 32,000 echoes limit for the apparatus. In fact, in these samples, a very mobile phase was observed that required more measurement points.

Before analysis, it was necessary to optimize the sample preparation procedure. After a series of preliminary experiments, the most appropriate way of sample preparation (shown in Figure 1) consisted of weighting precise amount of the MC sample into NMR glass tube (diameter 10 mm) and subsequent evacuation of the sample (Figure 1a). Demineralized water was poured, in known amounts, into a small glass container connected to an NMR tube with sample by valve (Figure 1b) and frozen by immersion in liquid nitrogen (−196 °C). The NMR tube, with the evacuated sample, was cooled in a container containing water and ice (0 °C, Figure 1c). The appropriate amount of water was added to the sample by opening the connecting valve between the NMR tube and the water-containing container. The several samples with different ratios of sample to water C:H_2_O (especially 1:2, 1:4, and 1:6, Figure 1d) were prepared in this way and labeled as MC-X:Y, where X:Y represents the mass ratio of C:H_2_O.

In addition to wetted samples (as described above), another set of samples stored in different controlled humidity atmospheres was also prepared. The prepared MC sample was dried in a vacuum oven at 150 °C for one night, then stored in a desiccator with silicagel (with almost 0% humidity) and denoted MC-0. Part of the MC-0 sample was removed and placed in another desiccator, with saturated solution of CaCl_2_ (anhydrous, 99.99%, Sigma Aldrich) and a relative humidity of 30%. The sample was stored for sufficient saturation under these conditions for 3 to 4 days (denoted MC-30). Another part of the MC-0 sample was placed in a desiccator with a demineralized water with relative humidity of nearly 100%. The sample was labeled as MC-100 and left under these conditions for 3–4 days. The content of water in the samples prepared by this procedure was determined by weighing. Table 1 summarizes the composition of all prepared samples for NMR relaxation measurements.

### 2.5. NMR Data Evaluation Procedure

The dependence of magnetization on time obtained (relaxation curve) was expected to be a linear combination of relaxation of the individual components differing in relaxation time *T*_2_. Thus, the measured relaxation curve can be expressed as [34]
(1)$$M=M_{0}+\sum_{i}A_{i}.e^{-\left(\frac{t}{T_{2,i}}\right)},$$
where *M* is the measured magnetization in time *t*, *A_i_* is the amplitude related to the amount of the *i*-th component, and *T*_2*,i*_ is its spin–spin relaxation time.

For a given number of components *i* expected to be present in the system, estimates of parameters *M*_0_, *A_i_*, and *T*_2*,i*_ were found by nonlinear optimization using the solver utility in Microsoft Excel 2016. The sum of squares of deviations between experimental and calculated data (RSC) was used as a criterion (to be minimized) to find the best estimates. The *A_i_* values were normalized after calculation ($\sum^{\;}A_{i}=100$).

For comparison, the magnetization decay (relaxation curve) was also analyzed using a numerical Laplace inversion algorithm in CONTIN (program integrated in Minispec software V3.00 Rev.10) to obtain the distribution of relaxation times *T*_2_.

## 3. Results and Discussion

### 3.1. Characterization of MC Sample by TPM and N_2_ Physisorption

The DSC record (curve of melting) of the MC-1:2 sample showed two clearly separated peaks (Figure 2a). The peak at temperature around −7 °C was attributed to the melting of water in the mesopores and was well-separated from the second one at higher temperature, which corresponded to the melting of bulk water. Evaluation of the obtained curve, according to reference [7], led to a quantitative description of the mesoporosity (Table 2), with the presence of pore diameters between ca. 5 and 20 nm (Figure 2b).

The mesoporous character of the MC sample studied was likewise confirmed on the basis of the shape of the nitrogen adsorption–desorption isotherm (Figure 3a), which corresponded to isotherm of IVa type, with a hysteresis loop of type H1 indicating a narrow range of mesopores [35]. Figure 3b illustrates the pore size distribution calculated by QSDFT theory, assuming slit-shaped micropores and cylindrical mesopores. The porosity of the sample was formed by a relatively narrow distribution of mesopores ranging from 8 to 13 nm and microporosity below 1.5 nm (Figure 3b). The porous characteristics of the sample derived from the pore size distribution, shown in Table 2, indicate that the sample was rather mesoporous (*V_meso_* = 0.98 cm^3^/g), rather than microporous (V_micro_ = 0.1 cm^3^/g). 

Both the TPM and N_2_ adsorption techniques undoubtedly show the mesoporous character of the MC sample and are in good agreement, regarding its high mesoporous volume.

### 3.2. ^1^H-NMR Relaxation Measurements

#### 3.2.1. Determination of Number of Components

The dried sample with almost 0% moisture (MC-0) and the sample stored in an atmosphere with 30% humidity (MC-30) both showed very fast relaxation, illustrated in Figure 4 as an example. Such a fast relaxation (below the detection limit of the used instrumentation) can be explained by the presence of only rapidly relaxing hydrogen nuclei, which are either part of a solid structure or adsorbed at favorable surface adsorption sites, such as oxygen-containing function groups. Thus, these samples are too dry to detect any relaxation of the hydrogen nuclei of water in pores or cavities.

To find a number of components in Equation (1) that well-fit the real relaxation behavior of the samples (except MC-0 and MC-30), we performed non-linear optimization of parameters *M*_0_, *A_i_*, and *T*_2*,i*_ (see Equation (1)), assuming a system composed of 1–5 components.

The procedure for searching the number of components is illustrated, for example, for the sample MC-1:6 in Table 3. Rows 1–5 in Table 3 indicate the increasing number of components used to optimize Equation (1). The experimental (empty circles) and calculated (solid circles) relaxation curves (second column in Table 3) illustrate that the increase in the number of components led to better agreement, with an improvement of the correlation coefficient *R*^2^ of up to three components. On the contrary, the analysis of residua (difference between experimental and calculated values, third column in Table 3) shows that even for the four components, the residua follow some trend, especially at short times, which indicates unsuitability of the model used. Only the model including five components leads to randomly distributed residua and can be evaluated as the best in a given case. As a result of optimization, the number of components, as well as their amplitudes *A* and relaxation times *T*_2_, were calculated (Table 3, fourth column).

This procedure was applied to all the samples/measurements studied, and the results (number of components and weighed average of relaxation times) are summarized in Table 4.

The sample stored in an atmosphere with 100% humidity (MC-100) exhibited the typical relaxation curve consistent with curves of more wet samples (MC-1:2, MC-1:4, and MC-1:6). The amount of water in sample MC-100 was more than ten times higher, compared to sample MC-30 (see Table 1), and allowed for the identification of two components with short relaxation times. Increasing the water content (sample MC-1:2) leads to an increasing number of components, and finally, the relaxation of the most wetted samples (MC-1:4 and MC-1:6) can be described by five components with long average relaxation times, which indicates the presence of bulk water outside porosity or in large interparticle cavities. Comparing the results for different values of time delay τ (0.05 or 0.2 ms) does not show any significant effect of this parameter.

#### 3.2.2. Analysis of Found Components

Based on the relaxation times determined for all components and the different samples, it appears that all *T*_2_ values can be divided into the following seven intervals, according to their similarity (<2 ms, 2–7 ms, 7–20 ms, 20–80 ms, 80–350 ms, 350–1500 ms, and above 1500 ms); these intervals are indicated in Figure 5. All found *T*_2*,i*_ of components from all measurements were collected for each *T*_2_ range and are listed in Table 5 with their corresponding normalized amplitude *A_i_* (%). If we assume that the found components represent nearly all of the present protons in water, the normalized amplitudes serve as the portion of water relaxing with given *T*_2_. Since the amount of water contained in the samples is known (Table 1), these *A_i_* amplitudes can be converted to the mass of water relaxing per gram of carbon (g H_2_O/g C) for each *T*_2*,i*_ population (Figure 6).

The differently wet samples showed varying relaxation of the proton nuclei (Table 5). Sample MC-100 contained the smallest amount of water, which filled the smallest pores (micropores) and covered the mesopores and external surfaces. Therefore, the water molecules were in strong interaction with the surface, and their relaxation was fastest with two components at 0.9 ms and 6.4 ms in this sample (Table 5).

As the amount of water increases, larger pores are expected to gradually fill up. For the MC-1:2 sample, in addition to fast relaxing water with *T*_2_ below 7 ms, the components with longer relaxation were observed, including a small amount of water relaxing longer than 1700 ms (which we expect to be bulk water). The MC-1:2 sample can be considered as a boundary between low-wet samples (MC-0, MC-30, and MC-100) with only partly filled porosity and well-wet samples (MC-1:4 and MC-1:6) with pores and cavities fully occupied with water. For these well-wetted samples (MC-1:4 and MC-1:6), the amount of water in the large pores and spaces relaxing approximately from 11 ms and up was too high, compared to rapidly relaxing water near the surface; thus, these fast-relaxing components could not be identified.

A better view of the distribution of individual components in the porous system is provided by recalculating their amplitudes to mass of water per g of carbon (Figure 6). 

The small amount of water (hydrogen nuclei), with very short relaxation times below 2 ms and 2–7 ms, was detected in less wetted samples MC-100 and MC-1:2. These regions of relaxation times should be connected with hydrogen atoms in very small pores (micropores) eventually in the surface layer of the material. The amount of water relaxing in this range was too small and could not be detected for other, wetter samples (MC-1:4 and MC-1:6). Another reason for the disappearance of the low *T_2_* values can be related to the interconnection of micropores with bigger pores [21].

The amount of relaxing water in the micropores should be around 0.1 g/g C because the volume of the micropores from the N_2_ adsorption was 0.1 cm^3^/g (Table 2). The amount of water in this region was slightly higher, about 0.2 g/g C (for MC-100) and 0.3 g/g C (for MC-1:2), which suggested completely filled micropores and some additional water (occurring in their vicinity or as a surface layer in larger pores). This comparison confirmed the prediction that relaxation times of up to 7 ms correspond to relaxation in micropores. 

The region around 7–20 ms should probably correspond to water in the mesopores. These relaxation times were recorded for all samples, except MC-100, which had the least amount of water. The pore volumes of the mesopores were 0.98 cm^3^/g (N_2_ adsorption) and 0.92 cm^3^/g (TPM). Totals of 1.1 g/g C (MC-1:2), 0.85 g/g C (MC-1:4 A), and 1.3 g/g C (MC-1:6) of water relaxing in regime 7–20 ms were observed. If we compare the amount of relaxing water in the mesopores with the pore volume of the mesopores from both techniques, we can reasonably mention good agreement between the results.

The other regions of relaxation times from 20 to 1500 ms are relatively widespread and can correspond to water in interparticle cavities. Relaxation times above 1500 ms evidently indicate bulk water surrounding the carbonaceous material. The relaxation time of bulk water is known to be about 3000 ms [36]. Long relaxation times (above 1500 ms) were recorded for sufficiently wetted samples, namely MC-1:4 and MC-1:6. 

To compare the results obtained with the simple calculation procedure described above, the experimental relaxation curves were analyzed by fitting analysis using the CONTIN algorithm, and the distribution of *T*_2_ with appropriate amplitudes was obtained. As illustrated in Figure 7a for the samples MC-1:4 and MC-1:6 (Figure 7b), the obtained *T*_2_ distribution consisted of five peaks at times corresponding to the intervals derived earlier (see Figure 6).

#### 3.2.3. Estimation of Surface Relaxivity

The conversion of the estimate of function *A(T)* as a distribution of water, according to the relaxation time to pore size distribution, is based on the relation [37]
(2)$$\frac{1}{T_{2}}=\frac{1}{T_{2B}}+\rho \cdot \frac{S}{V}\;,$$
where *T*_2_ is the relaxation time of the water in the pores, *T*_2*B*_ is the relaxation time of the bulk water, *ρ* is the surface relaxivity for spin–spin relaxation, *S* is the surface, and *V* is the pore volume. 

If we expect cylindrical pores (the same pore geometry as for N_2_ physisorption and TPM) in the studied material, then
(3)$$\frac{S}{V}=\frac{2}{r}=\frac{4}{d},$$
where *r* and 20*d* are the pore radius and diameter, respectively.

The combination of Equations (2) and (3) lead, after rearrangement, to a form allowing for recalculation of relaxation time *T*_2_ to the diameter of the pores:(4)$$d=\frac{4\rho T_{2B}T_{2}}{T_{2B}-T_{2}}$$

Considering the fact that *T*_2*B*_ >> *T*_2_, the relation can even be simplified:(5)$$d=4\rho T_{2}$$

The restriction of Equation (5) lies in the value of surface relaxivity *ρ*. This parameter can be obtained by measuring standards with known (meso)pore size and shape or by NMR diffusometry and other methods [21,38,39]. Another limitation that affects the use of this equation is the dependence of the relaxivity on surface chemistry. Due to the absence in the literature of relaxivity values for carbonaceous materials, we estimate ρ  using two different approaches. 

The first one is based on Equation (2), which can be simplified and rewritten as
(6)$$\rho =\frac{V}{S\cdot T_{2}}$$
considering that *T*_2*B*_ >> *T*_2_ and *T*_2_ being here, in the absence of paramagnetic centers, the transverse relaxation time of the water molecule layer bonded to the surface, and in fast exchange mode, with the surrounding water. Based on the pore size distribution determined by N_2_ physisorption data, three types of pores have been evidenced: ultramicropores (pore size < 0.7 nm), supermicropores (2 > pore size > 0.7), and mesopores are present. Each type of pore contributes to the relaxation rate; the smaller the pore, the faster the relaxation (and the lower the *T*_2_ value). Therefore, it was proposed to attribute the *T*_2_ population at 0.9 ms to water molecules tightly adsorbed in ultramicropores. The restricted motion of these molecules and the poor accessibility of this type of pore strongly hinder its exchange with other water molecule and this *T*_2_ was not considered for the calculation of the relaxivity. The *T*_2_ population at 6.2 ms was then attributed to the exchangeable water layer located on the remaining surface formed by the supermicropores and mesopores. Indeed, for mesoporous carbon obtained by the soft-template process, the microporosity is located in the thin carbon walls (a few nanometers thick), with a direct connection to the mesopores. Based on the value of Table 1, the surface relaxivity *ρ* was then estimated at 0.35 nm/ms, considering *V* (and *S*) as the sum of the supermicroporous and mesoporous contributions of *V* (and *S*). Then, the value of the diameter of the mesopores (from Equation (5)) correspond to 16 nm. This value is in order of magnitude of the diameter of the mesopores determined by the other techniques (see Table 2). 

The second approach to determine the *ρ* is based on the use of the diameter of mesopores obtained by other techniques (14 nm by TPM analysis and 9 nm by N_2_ adsorption), with the averaged value of the relaxation time *T*_2_ (11.7 ms) corresponding to the water in the mesopores. The calculation led to relaxivity values of 0.30 nm/ms and 0.19 nm/ms, respectively. 

The surface relaxivities calculated by both approaches are reasonably comparable to the relaxivity of organic materials (e.g., 0.44 nm/ms for porous polymer particles [21]), considering that a significant amount of remaining organic fraction is present for the MC sample, due to its low pyrolysis temperature (400 °C). 

## 4. Conclusions

The porosity of a different wetted carbonaceous sample was investigated by low-field ^1^H NMR relaxometry. The porosity of the sample was also analyzed by more usual techniques, such as N_2_ physisorption and thermoporometry (TPM). On the basis of the aim of our study, we may draw the following conclusions. 

We present a new method for the conversion of relaxation curves to the distribution of relaxation times *(T*_2_*)*, without the need for special software. The procedure consists of two simple steps: (i) determination of the number of ^1^H different nuclei (components) of water molecules by the nonlinear optimization of parameters in Equation (1) using the widely available solver utility in Microsoft Excel, together with the analysis of residues. (ii) Distribution of components into groups of relaxation times corresponding to different spaces (pore sizes) filled in the sample. 

We have revealed 2–4 different components for the least wet samples and 5 components for the most wet sample. The shortest *T*_2_ (< 7 ms) were recorded for samples containing the smallest amount of water, which was attributed to the surface layer and water in the micropores. As the amount of water increased, larger pores gradually filled. Nitrogen physisorption and TPM revealed the presence of 8–13 nm pores and 5–20 nm pores, respectively, which, for well-wetted samples, corresponds to relaxation times of *T*_2_ = 7–20 ms. The interparticle cavities were attributed to the *T*_2_ region = 20–1500 ms. Higher *T*_2_ values are attributed to the bulk water surrounding the carbonaceous material. 

Interestingly, a narrow range of surface relaxivity values was presented for the first time for carbon materials. These values were calculated using two different approaches. The first one is based on the total surface and total pore volume (excluding ultramicropores) determined by N_2_ adsorption and is based on the *T*_2_ components attributed to the mesopores (11 ms) and to the surface bounded water (6 ms). It allows for the calculation of the mesopore value in the wet state (16 nm), in close agreement with the value determined by TPM in the wet state, as well. The second approach is based on the use of the mesopore diameters known from TPM or gas physisorption data and the *T*_2_ value for water confined in mesopores. 

This contribution can serve as a stepping stone for further research, especially toward the estimation of pore size distribution of carbon materials by simple ^1^H NMR relaxometry. Further work should be focused on the extension of the proposed data treatment and surface relaxivity determination of carbonaceous materials, including the conversion of distribution of relaxation times, pore size distribution, and wider variety of the sample. Firstly, the set of carbon materials differing in pore size, shape, hierarchy, connectivity, and degree of ordering, together with analogous surface chemistry, has to be studied under comparable measurement conditions. Secondly, the results should be compared to ^1^H NMR cryoporometry or diffusometry to obtain the full image of behavior and relaxation of protons in water confined in carbonaceous porous systems. 

## Figures and Tables

**Figure 1 materials-15-09021-f001:**
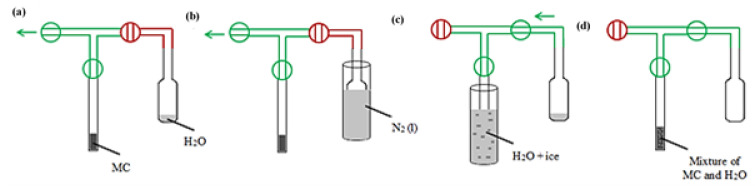
Scheme of preparation of MC sample for NMR analysis: (**a**) evacuation of the dry sample, (**b**) freezing the reservoir with water, (**c**) sublimation of water to sample, (**d**) final prepared sample.

**Figure 2 materials-15-09021-f002:**
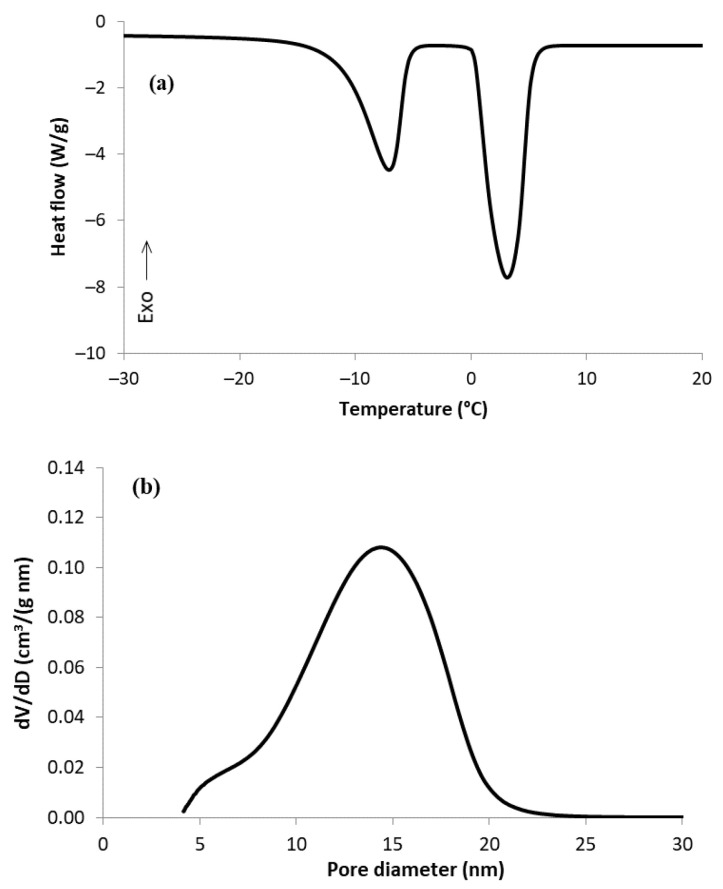
DSC curve of melting (**a**) and pore size distribution from TPM (**b**) of MC-1:2 sample.

**Figure 3 materials-15-09021-f003:**
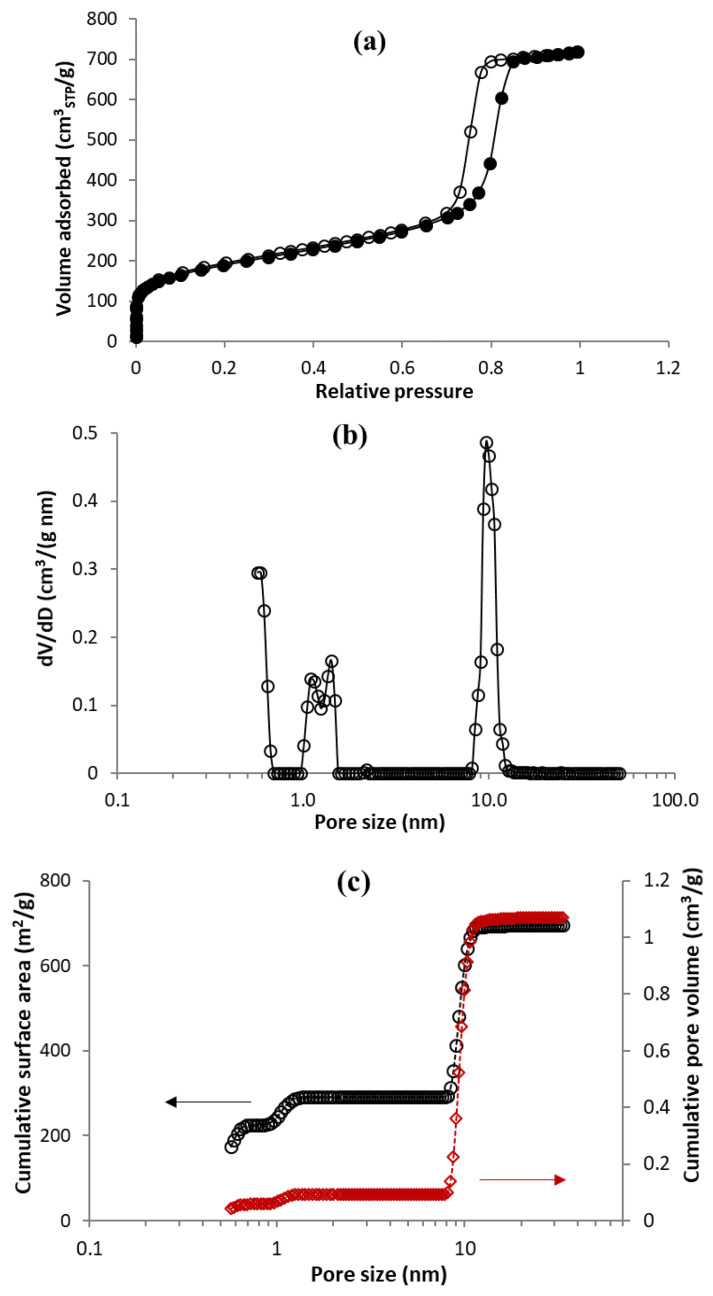
Nitrogen adsorption at −196 °C (solid circles) and desorption (empty circles) isotherms on MC sample (**a**), semi-logarithmic pore size distribution obtained by DFT method (**b**), and corresponding cumulated surface area (black) and pore volume (red) (**c**).

**Figure 4 materials-15-09021-f004:**
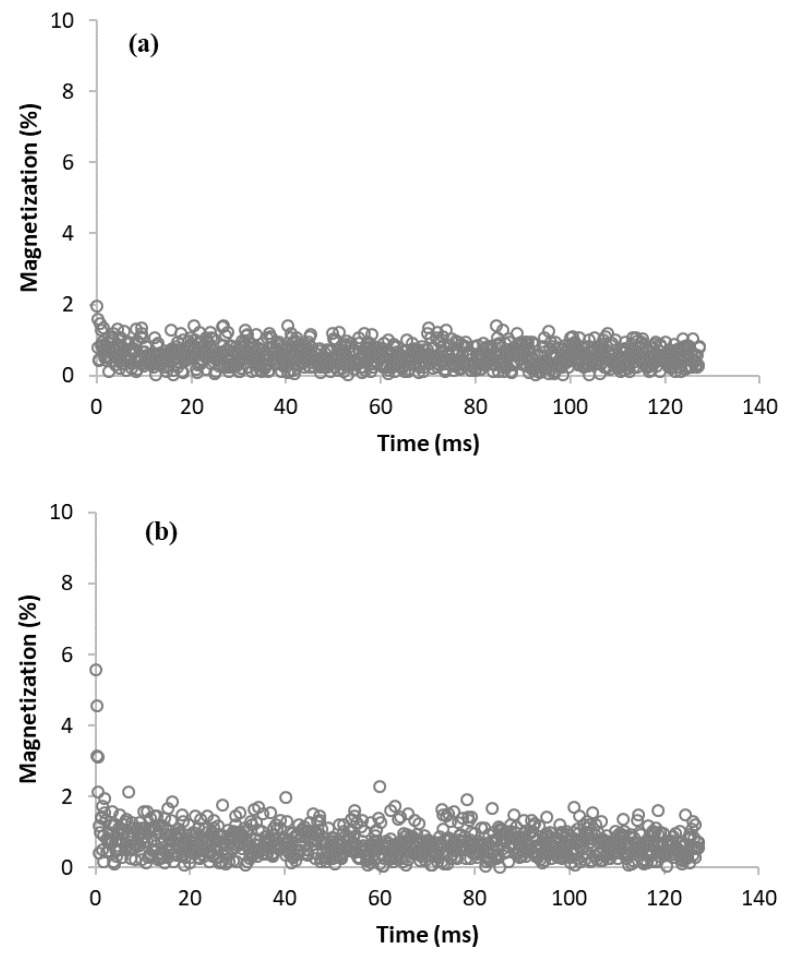
Relaxation curves of dried sample (MC-0) (**a**), and sample stored in atmosphere with 30% humidity (MC-30) (**b**).

**Figure 5 materials-15-09021-f005:**
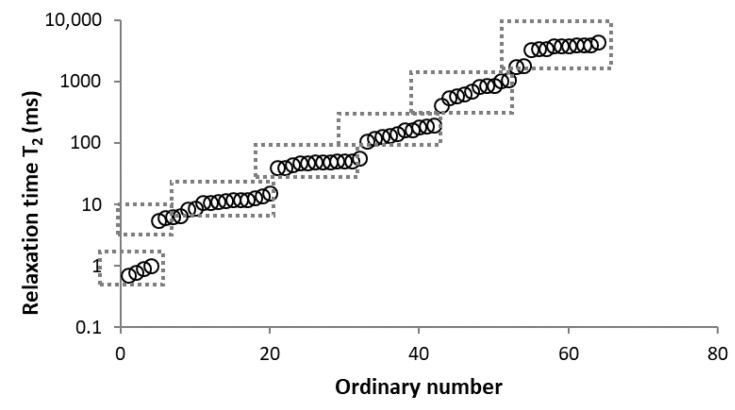
Grouping of relaxation times values obtained on the different samples (logarithmic scale on vertical axis) into different intervals.

**Figure 6 materials-15-09021-f006:**
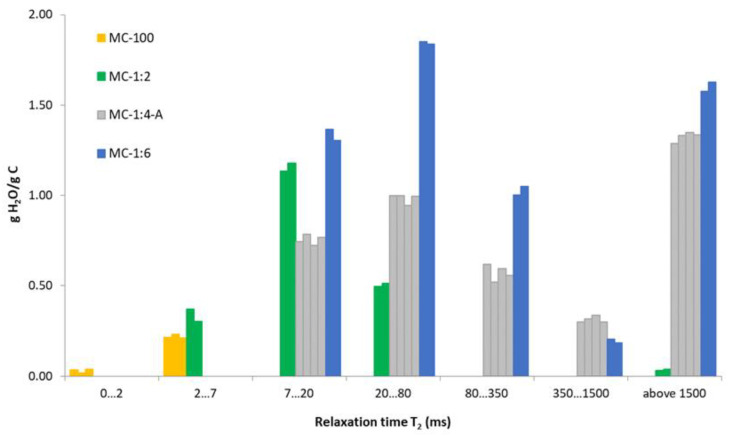
Content of water nuclei calculated from amplitudes and relaxing for each appropriate interval of *T*_2_ relaxation time.

**Figure 7 materials-15-09021-f007:**
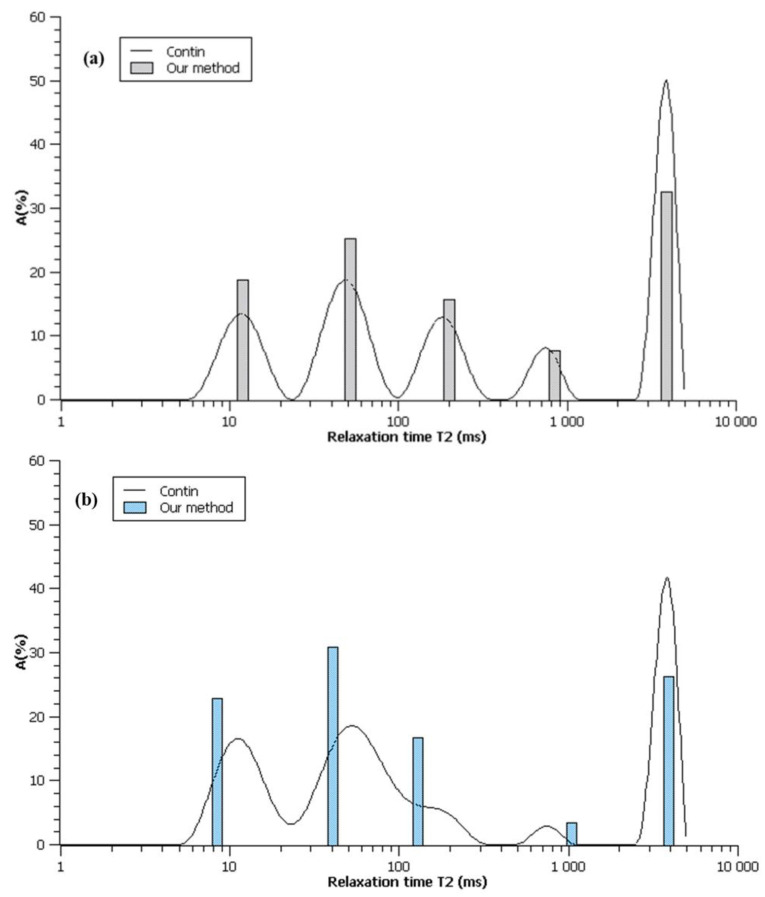
Distribution of *T*_2_ for MC-1:4 (**a**) and MC-1:6 (**b**) by CONTIN algorithm, compared to our results (logarithmic scale on horizontal axis).

**Table 1 materials-15-09021-t001:** Preparation parameters of all investigated samples.

Sample	Mass of Dry Sample (g)	Mass of H_2_O (g)	Mass Ratio H_2_O/C
MC-0-A	0.1368	0	0
MC-0-B	0.1407	0	0
MC-30-A	0.1108	0.0024	0.02
MC-30-B	0.1142	0.0024	0.02
MC-100-A	0.1536	0.0391	0.25
MC-100-B	0.1520	0.0387	0.25
MC-1:2	0.1451	0.2939	2.03
MC-1:4-A	0.1099	0.4341	3.95
MC-1:4-B	0.1173	0.4643	3.96
MC-1:4-C	0.1080	0.4342	4.02
MC-1:6	0.0899	0.5393	6.00

(A, B, C—correspond to repeated sample preparation).

**Table 2 materials-15-09021-t002:** Porous characteristics of MC sample determined by TPM and N_2_ adsorption at −196 °C.

Method	*V_ultramicro_* (cm^3^/g)	*S_ultramicro_* (m^2^/g)	*V_supermicro_* (cm^3^/g)	*S_supermicro_* (m^2^/g)	*V_meso_* (cm^3^/g)	*S_meso_* (m^2^/g)	Mesopore Diameter (*) (nm)
TPM	n/a	n/a	n/a	n/a	0.92	n/a	14
N_2_ adsorption	0.06	225	0.04	65	0.98	405	9

(*) the most frequent.

**Table 3 materials-15-09021-t003:** Illustration of searching of components number for sample MC-1:6. The experimental (empty circles) and calculated (solid circles) relaxation curves. Black curve in residues represent weighed average of residues.

Number of Comp.	Relaxation Curve	Residues	Calculated Parameters	
			A (%)	*T*_2_ (ms)
1	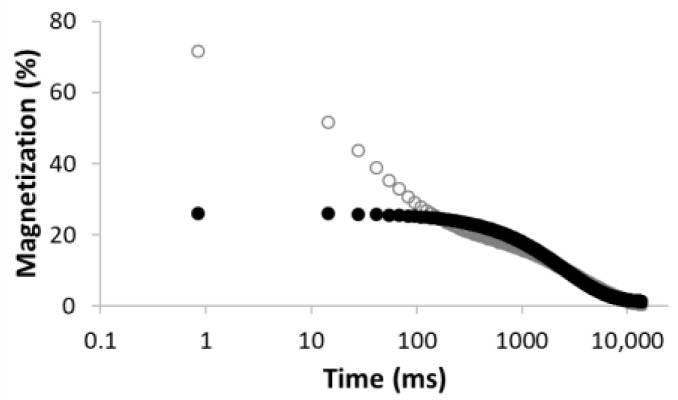 R^2^ = 0.888	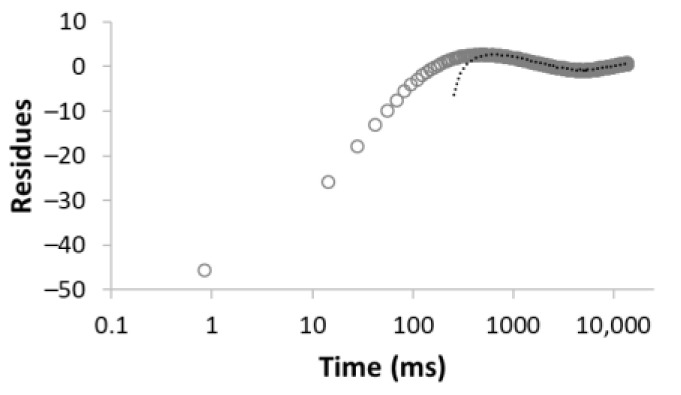	100	2495.5
2	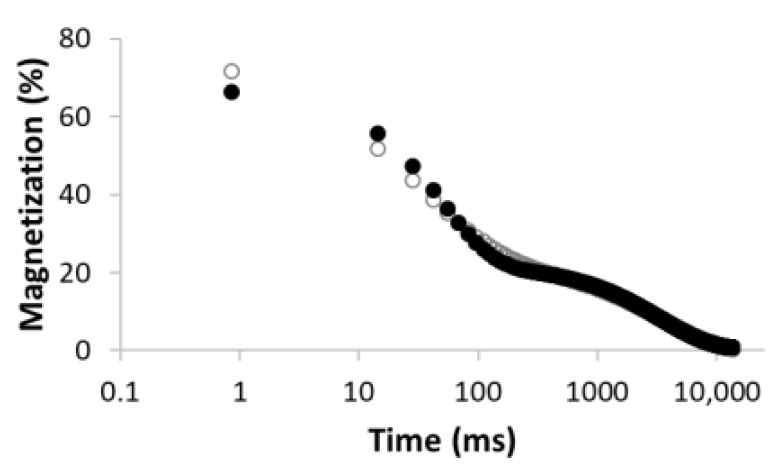 R^2^ = 0.996	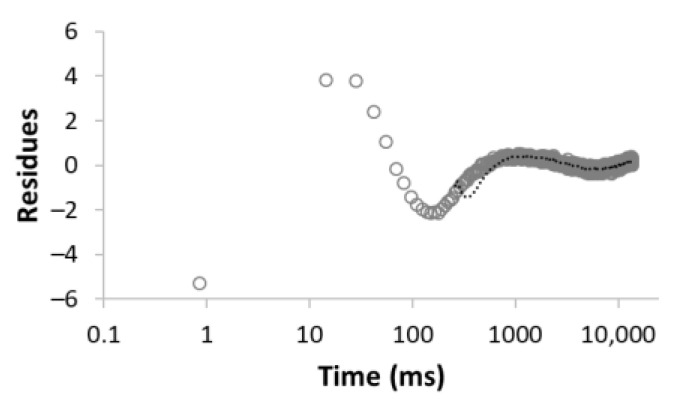	67.8 32.2	49.0 3320.5
3	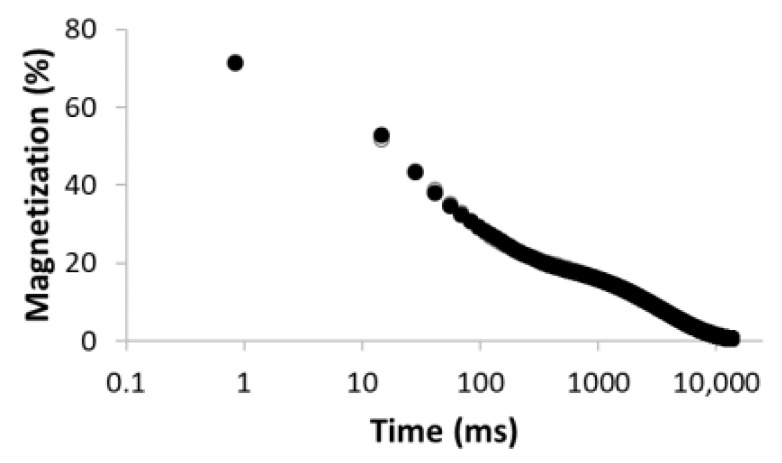 R^2^ = 0.999	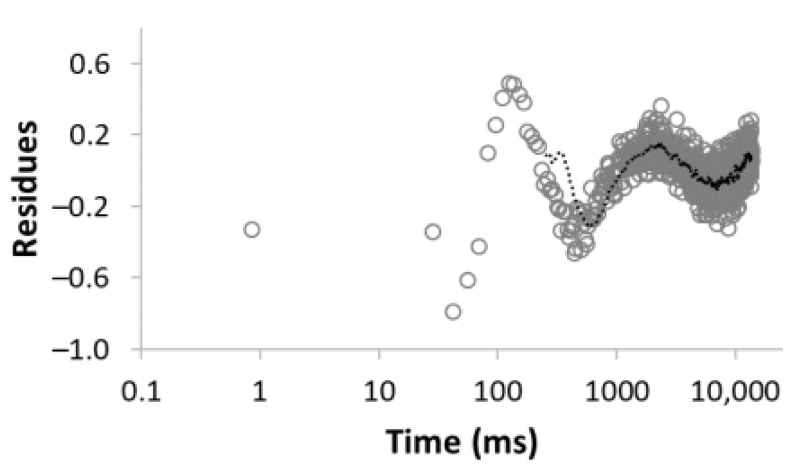	43.4 27.9 28.6	17.5 113.9 3534.6
4	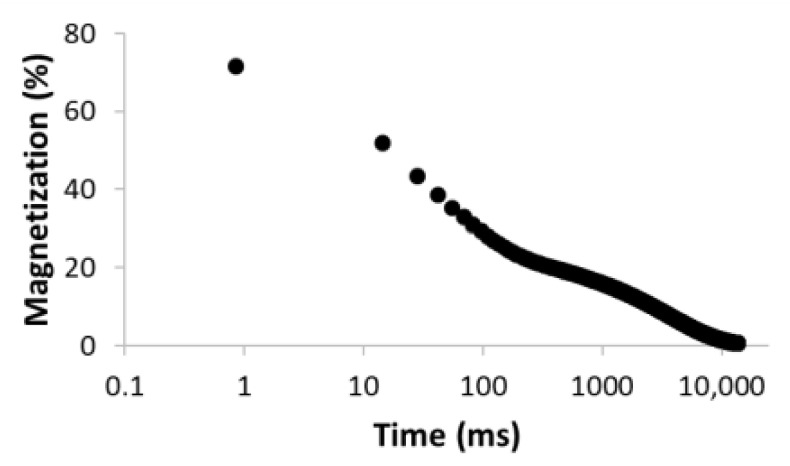 R^2^ = 0.999	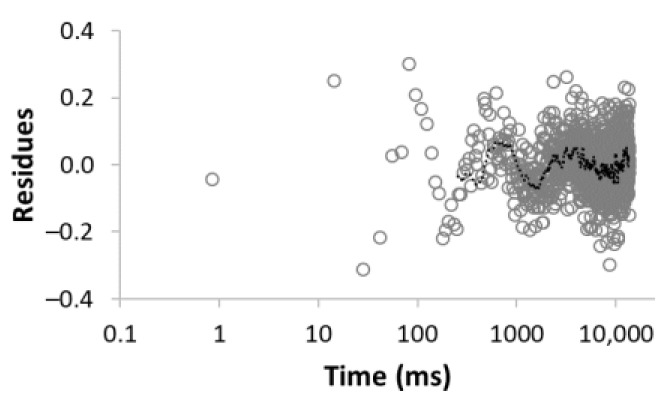	32.2 69.5 5.2 27.5	11.9 35.1 429.5 3711.6
5	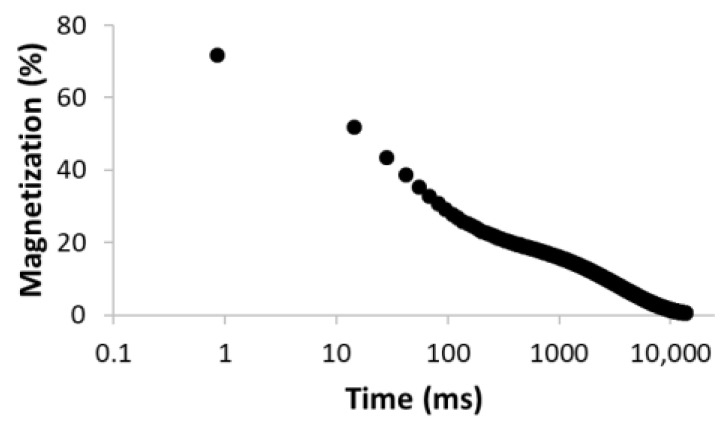 R^2^ = 0.999	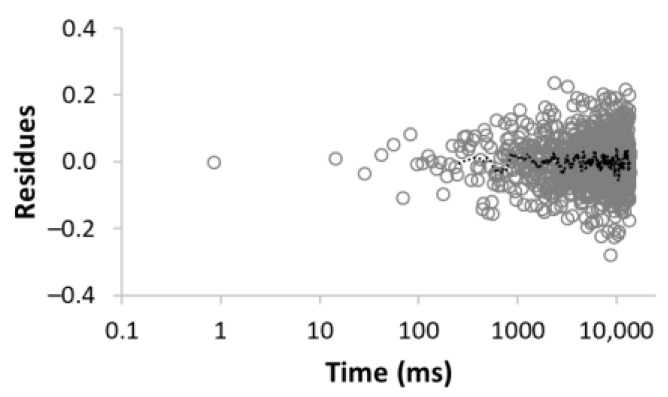	22.8 30.8 16.7 3.4 26.3	8.4 39.9 128.2 1028.2 3861.2

**Table 4 materials-15-09021-t004:** Optimized number of components determined for the fitting of the magnetization decay and for corresponding weighed average of relaxation times. Measurements were repeated, and each experiment was given numbers put in brackets.

Sample (Measurement)	τ (ms)	Number of Components	*T*_2_ (ms)
MC-100 (1)	0.05	2	5.5
MC-100 (2)	0.05	2	6.2
MC-100 (3)	0.05	2	5.5
MC-1:2 (1)	0.2	3	20.2
MC-1:2 (2)	0.05	4	22.4
MC-1:4-A (1)	0.2	5	1353
MC-1:4-A (2)	0.2	5	1228
MC-1:4-A (3)	0.05	5	1386
MC-1:4-A (4)	0.05	5	1413
MC-1:4-B (1)	0.2	5	1845
MC-1:4-B (2)	0.05	5	1620
MC-1:4-C (1)	0.2	5	1299
MC-1:4-C (2)	0.05	5	1209
MC-1:6 (1)	0.2	5	1086
MC-1:6 (2)	0.05	5	1129

**Table 5 materials-15-09021-t005:** Values of relaxation times (*T*_2_) and amplitudes (*A*) for all studied samples belonging to appropriate intervals.

Interval *T*_2_ (ms)	<2		2–7		7–20		20–80		80–350		350–1500		Above 1500	
Sample	A (%)	*T*_2_ (ms)	A (%)	*T*_2_ (ms)	A (%)	*T*_2_ (ms)	A (%)	*T*_2_ (ms)	A (%)	*T*_2_ (ms)	A (%)	*T*_2_ (ms)	A (%)	*T*_2_ (ms)
MC-100 (1)	14.1	0.9	85.9	6.2										
MC-100 (2)	7.2	0.7	92.8	6.6										
MC-100 (3)	15.2	1.0	84.8	6.3										
MC-1:2 (1)	18.2	0.8			55.9	13.9	24.4	50.3					1.5	1781
MC-1:2 (2)			14.9	5.6	58.0	15.4	25.3	49.8					1.8	1758
MC-1:4-A (1)					18.8	11.8	25.3	50.8	15.7	195.6	7.6	823	32.6	3818
MC-1:4-A (2)					19.8	11.7	25.3	51.1	13.2	188.1	8.0	874	33.7	3318
MC-1:4-A (3)					18.4	10.9	23.9	47.4	15.0	165.1	8.5	592	34.1	3806
MC-1:4-A (4)					19.4	11.3	25.2	48.0	14.1	162.1	7.5	697	33.8	3916
MC-1:4-B (1)					19.5	13.0	27.4	56.9	7.7	182.6	5.3	1051	40.1	4382
MC-1:4-B (2)					17.8	10.6	22.1	43.8	14.3	105.4	4.1	410	41.6	3789
MC-1:4-C (1)					19.1	12.1	23.4	48.9	17.6	139.8	4.3	852	35.5	3449
MC-1:4-C (2)					19.9	12.1	23.4	49.0	17.7	132.4	5.3	551	33.7	3390
MC-1:6 (1)					22.8	8.4	30.8	39.9	16.7	128.2	3.4	1028	26.3	3861
MC-1:6 (2)					21.7	8.7	30.6	39.9	17.5	117.3	3.1	633	27.1	3965
**Average**		0.9		6.2		11.7		48.0		151.7		751		3436

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
