# Peer review of "Carbonaceous Materials Porosity Investigation in a Wet State by Low-Field NMR Relaxometry"

_materials, 2022, doi:10.3390/ma15249021_

Round 1
Reviewer 1 Report
This paper describes an investigation of wetted porous carbon materials using NMR relaxometry, nitrogen adsorption and DSC techniques. A simplified procedure for assessing pore size distribution is given.
The relaxation times enable different pore sizes to be distinguished. The results from the simplified analysis are comparable with a Laplace analysis using the CONTIN algorithm. And pore sizes in good agreement with N2 adsorption.
Since the method is already well-established, the main novelty of the work is the much simplified analysis procedure.
The paper is clearly written, with full details of the results, materials and procedures, and is suitable for publication in its present form.
Reviewer 2 Report
The authors of the manuscript Carbonaceous materials porosity investigation in a wet state by low-field NMR relaxometry describe the possibility of evaluating the porosity of adsorbents using low-field 1H NMR relaxometry.
The methods used are well described, and the presented results are justified and well argued.
My opinion is that the presented manuscript can be published in its present form.
Reviewer 3 Report
Dear Authors,
a potentially interesting manuscript, but with significant gaps to be considered complete.
It will indicate the most important issues for improvement in my opinion.
1. Better discussion in the context of what has been done in this field (so far), I recommend here the analysis and discussion of issues present in the literature in the context of LF-NMR:
- low field 1H NMR characterization of mesoporous silica MCM-41 and SBA-15
- Identification of Proton Populations,
- overcoming the barriers of nanoporous porosity.
2. Description of possible methods of quality verification of the proposed method. Here, the minimum requirement is to describe (and discuss the importance of) further steps to verify the method based on research, complementary measurements, e.g. using phantoms or other LF-NMR techniques and approaches.
Some suggestions:
Analyzing the issues listed in point 1, the following topics arise for clarification and discussion:
- TE time = 100us, for many nanopore systems it is insufficient to measure T2 from all protons filling the pores, here a form of calibration to the standard proton content, e.g. H2O, is recommended, then calibration of porosity recorded by T2 versus e.g. assessment of correctness of measurements. And here the amplitude is insufficient, the integral under the T2 curve (normalized to the standard) is needed. Additionally what about the proton signal from the probe (what about coil ringing?) and protons that can bind to carbon, shouldn't that be subtracted before the ILT transform? It seems that for the analysis of e.g. ILT a differential signal should be taken?
- the authors do not take into account the effect of diffusion on the distribution of PSD by T2, below 10 nm it should be noticeable.
- why MICP was not additionally taken into account, then taking into account the physical limitations for this 3-5 nm technique, we more precisely set our PSD in the nanometer range (with a large share of pores of several nanometers)
- why we don't see the measurement for TE=400 us mentioned by the authors
- what is the SNR for TE=100us, how much was the accumulation, the signal looks very noisy
These and other teams should be discussed in the context of the given literature issues.
Round 2
Reviewer 3 Report
Dear Authors,
Your answers partially improve the manuscript. I understand that it is difficult to correct experimental work now. Therefore, I suggest an even more careful discussion with the results of nanometric objects available in the literature (the aforementioned "overcoming" of shale rock cores with a dominant pore diameter of 2 nm, nanometric structure containing organic matter, i.e. carbon structures ...; as well as clay materials, such as illite, smectite, kaolinite, chlorite, kerogen,.., all measured by LF-NMR and complementary methods).
The reference to the MICP issue is a practical suggestion for the future that the combination:
LF-NMR, N2 and MICP, and additionally mass-volume, as complementary techniques, allows us to better determine where we actually are in our PSD (Pore size distribution) built largely of nanometric pores.
In the suggested issues, these approaches are explored more extensively.
In addition, I suggest recalculating the given value of surface relaxitvity, in my opinion it is ~ 0.3 um/s, etc., not nm/s.
In order not to prolong the process of publishing the manuscript, which is quite interesting, I am giving a minor revision. However, I suggest a more extensive discussion with the results available in the literature, in particular those containing LF-NMR with TE times of 100 and 60 µs.
I look forward to continuing the work.
